# Characterization of the Genomic Landscape in Cervical Cancer by Next Generation Sequencing

**DOI:** 10.3390/genes13020287

**Published:** 2022-01-31

**Authors:** Ling Qiu, Hao Feng, Hailin Yu, Ming Li, Yana You, Shurong Zhu, Wenting Yang, Hua Jiang, Xin Wu

**Affiliations:** 1Department of Gynecological Oncology, Obstetrics and Gynecology Hospital, Fudan University, Shanghai 200082, China; QiuLingpang@163.com (L.Q.); fenghao_o@126.com (H.F.); slina_2002@163.com (H.Y.); familybmgwforever@163.com (M.L.); you90169025@163.com (Y.Y.); fanta8023@163.com (S.Z.); 2Department of Genetics, Stanford University School of Medicine, Stanford, CA 94305, USA; wenting@stanford.edu

**Keywords:** cervical cancer, genetic traits, therapeutic target, multigene NGS panel, PIK3CA

## Abstract

Cervical cancer is the fourth leading cause of cancer-related deaths in women worldwide. Although many sequencing studies have been carried out, the genetic characteristics of cervical cancer remain to be fully elucidated, especially in the Asian population. Herein, we investigated the genetic landscape of Chinese cervical cancer patients using a validated multigene next generation sequencing (NGS) panel. We analyzed 64 samples, consisting of 32 tumors and 32 blood samples from 32 Chinese cervical cancer patients by performing multigene NGS with a panel targeting 571 cancer-related genes. A total of 810 somatic variants, 2730 germline mutations and 701 copy number variations (CNVs) were identified. FAT1, HLA-B, PIK3CA, MTOR, KMT2D and ZFHX3 were the most mutated genes. Further, PIK3CA, BRCA1, BRCA2, ATM and TP53 gene loci had a higher frequency of CNVs. Moreover, the role of PIK3CA in cervical cancer was further highlighted by comparing with the ONCOKB database, especially for E545K and E542K, which were reported to confer radioresistance to cervical cancer. Notably, analysis of potential therapeutic targets suggested that cervical cancer patients could benefit from PARP inhibitors. This multigene NGS analysis revealed several novel genetic alterations in Chinese patients with cervical cancer and highlighted the role of PIK3CA in cervical cancer. Overall, this study showed that genetic variations not only affect the genetic susceptibility of cervical cancer, but also influence the resistance of cervical cancer to radiotherapy, but further studies involving a larger patient population should be undertaken to validate these findings.

## 1. Introduction

Cervical cancer remains a prevalent disease globally. Despite the introduction of screening and vaccination programs, there were approximately 570,000 new cases and 311,000 deaths from cervical cancer worldwide in 2018, making it the fourth most frequently diagnosed cancer and the fourth leading contributor to cancer-related mortality in women [1]. The high-risk factors of cervical cancer chiefly include human papillomavirus (HPV) infection, initiation of sexual behavior at a young age, multiple sexual partners, smoking, and long-term consumption of oral contraceptives [2]. Moreover, it has been reported that during carcinogenesis of cervical cancer, HPV DNA is frequently integrated into the human genome. Although the combination of surgery and radiochemotherapy has improved overall survival (OS), progression-free survival (PFS) and disease-free survival of cervical cancer patients, and reduced the recurrence rate of cervical cancer, the 5-year survival rate of advanced cervical cancer patients, especially metastatic cervical cancer patients, remain dismally low, ranging from 5% to 15% [3]. Moreover, the incidence and mortality of cervical cancer tend to be higher in countries or regions with a low human development index; cervical cancer is the most frequently occurring cancer type in women in sub-Saharan Africa and Southeast Asia [1]. Remarkably, China contributed more than a sixth of the global cervical cancer burden, with 106,000 new cases and 48,000 deaths in 2018 [4]. Consequently, greater efforts are needed to further elucidate the molecular mechanisms underlying tumor initiation and progression, especially in Chinese cervical cancer patients, which could facilitate the discovery of novel biomarkers for early cervical cancer screening and better molecular targets for the treatment of cervical cancer.

Over the recent years, many genomics sequencing studies have been carried out to uncover specific gene variations of cervical cancer. Kyrgiou et al., undertook a genome-wide association study (GWAS) of 273,377 women, including 4769 cervical intraepithelial neoplasia (CIN) grade 3 or invasive cervical cancer patients, and showed that six independent genetic susceptibility variants, PAX8 (rs10175462), CLPTM1L (rs27069), HLA-DQA1 (rs9272050), MICA (rs6938453), HLA-DQB1 (rs55986091) and HLA-B (rs92666183), were associated with CIN3 and invasive cervical cancer, suggesting disruptions in apoptotic and immune function pathways [5]. Yang et al., found that targeting β-catenin reverses radioresistance of cervical cancer carrying PIK3CA-E545K, the most common hotspot mutation of PIK3CA in cervical cancer [6]. Zhang et al., identified and screened the key genes (such as TSPO, CCND1) and pathways (such as DNA replication, organelle fission, chromosome segregation and cell cycle phase transition) closely related to cervical cancer by reanalyzing cervical cancer-associated gene expression dataset including 10 normal cervix samples and 21 cervical cancer samples [7]. Burk et al., identified SHKBP1, ERBB3, CASP8, HLA-A and TGFBR2 as significantly mutated genes and unraveled amplifications in BCAR4, CD274 and PDCD1LG2 in 228 primary cervical cancer, among which multiple genes can be used as therapeutic targets [8].

Although increasing cervical cancer-related mutations have been uncovered, the pathogenesis of cervical cancer remains still unclear in a considerable proportion of patients, and data are especially limited on the genetic characteristics of Chinese cervical cancer patients. In the current study, we used a multigene next generation sequencing (NGS) panel to analyze the sequencing results of 32 cervical cancer samples and paired normal control samples from Chinese cervical cancer patients. The panel contains 571 validated tumor-related genes and includes multiple genetic tests for simultaneously identifying single nucleotide variants (SNVs), small insertions and deletions (indels), copy number variations (CNVs), splice variants and gene rearrangements. We uncovered frequent and novel genetic alterations and performed related signaling pathways enrichment analysis, revealing distinct mutation characteristics from Caucasian patients.

## 2. Materials and Methods

### 2.1. Cervical Cancer Patients and Tissue Cohort

This study carried out between March 2019 and March 2020 at the Obstetrics and Gynecology Hospital, Fudan University, Shanghai, China, prospectively enrolled 32 consecutive patients with pathologically proven primary cervical cancer for ultradeep NGS using a 571-gene targeted sequencing panel. All the specimens were collected during surgical resection of the primary tumor. Data were acquired from the hospital’s electronic medical records system and by direct interviews of participants. Cervical cancer surgical samples, paired tumor tissue and blood samples were acquired at the Department of Gynecological Oncology of the Hospital.

### 2.2. Genomic DNA Isolation and Targeted NGS

All samples were processed in a next-generation sequencing laboratory (Xinshu Healthcare Technology Company, Shanghai, China). Library was prepared according to the instructions of each manufacturer. Genomic DNA was extracted using a QIAamp DNA Mini kit (Qiagen GmbH, Dusseldorf, Germany). The quantity and purity of DNA were assessed using a Qubit^®^ 3.0 fluorometer (Invitrogen; Thermo Fisher Scientific, Singapore) and a NanoDrop ND-1000 (Thermo Fisher Scientific, Wilmington, NC, USA). DNA fragmentation was evaluated by Genomic DNA ScreenTape assays (Agilent Technologies, Santa Clara, CA, USA) using the Agilent 2200 TapeStation system to produce a DNA integrity number. Sheared genomic DNA was used to perform end repair, A-tailing and adapter ligation with a KAPA library preparation kit (Kapa Biosystems, Wilmington, NC, USA). Libraries were captured using Agilent SureSelect human exon probes and amplified. Finally, the constructed sample libraries were sequenced by Illumina NextSeq500 System (Illumina, San Diego, CA, USA).

### 2.3. Preprocessing of Sequencing Reads

Raw short sequence reads were trimmed and filtered by fastp. Clean reads were mapped to the human reference genome hg19 using BWA-MEM with default parameters. Following GATK4 best practice, PCR duplicates in BAM files were first removed and subsequently realigned and recalibrated.

### 2.4. Somatic Variant Identification

Somatic SNVs and indels were identified using MuTect2. Tumor samples were used to call somatic mutations against the paired normal samples. Artifacts were filtered using the GATK FilterMutectCalls tool. Filtered variants were annotated using SnpEff with ExAC, 1000G, dbsnp, clinvar and COSMIC databases. The average depth of the sequencing was 1814X. To filter out mutations that may be false positive, only those mutations with a sequencing depth larger than 10X and supported by at least four mutation reads with a variant allele frequency (VAF) >0.01 and a global frequency <0.05 in ExAC and 1000G were used for further analysis.

### 2.5. Germline Variant Identification

Germline SNVs and indels were identified from the bam data of the blood samples using GATK HaplotypeCaller. Filtered variants were annotated using SnpEff with ExAC, 1000G, dbsnp, clinvar and COSMIC databases. To filter out mutations that might be false positive, only those mutations with a sequencing depth larger than 20X and supported by at least 10 mutation reads with a VAF >0.1 and a global frequency <0.05 in ExAC and 1000G were used for further analysis.

### 2.6. Copy Number Variations (CNVs)

CNVs were determined using CNVkit. A copy number of 1 indicated copy number loss, 0 homozygous deletion and ≥3 copy gain. ABSOLUTE was used to estimate tumor purity and ploidy from CNV and SNV results. Then, CNV was corrected for tumor purity and ploidy. In addition, significantly recurrent focal genomic regions with somatic copy number alterations (SCNAs) that were gained or lost in cervical cancer samples were identified using the Genomic Identification of Significant Targets in Cancer (GISTIC 2.0) algorithm20 software) [9]. Default parameters of GISTIC were used and focal events with q-value below 0.25 were considered as significantly recurrent. Significant focal events in individual samples were classified according to the amplitude threshold of GISTIC: GISTIC status = 0, below threshold; GISTIC status = 1, amplified (gain); GISTIC status = 2, highly amplified (amplification); GISTIC status = −1, deleted (loss); GISTIC status = −2, highly deleted (deletion). The rates of CNVs and SCNAs in early and advanced stage cervical cancer were analyzed.

### 2.7. Gene Ontology (GO) and KEGG Pathway Enrichment Analyses

Gene ontology analysis (GO) is commonly used for annotating large scale genes and gene products [10,11]. KEGG is a collection of databases dealing with genomes, diseases, biological pathways, drugs and chemical materials [12]. It is generated by molecular level information, can be used to predict which pathways a particular gene is enriched. It covers information resources such as diseases and pathways. GO analysis and KEGG analysis were performed by DAVID tools (DAVID. Available online: https://david.ncifcrf.gov/, accessed on 10 December 2021) [13] Statistical significance was considered for *p* < 0.01. DAVID, which is an online bioinformatic tool, is designed to identify a large number of genes or proteins function. We could use DAVID to visualize the DEGs enrichment of BP, MF, CC and pathways (*p* < 0.05).

### 2.8. Protein Interaction Assay

Protein interaction enrichment was analyzed using Metascape (Metascape. Available online: https://metascape.org/, accessed on 10 December 2021) [14]. The protein networks constructed were based on physical interactions among all input protein (gene) candidates.

## 3. Results

### 3.1. Patient Characteristics

The study included 32 paired cervical cancer and normal control samples from 32 consecutive patients. Twenty-four patients had squamous cell carcinoma, 4 patients had adenosquamous carcinoma, 2 patients had adenocarcinoma, and 1 patient each had endometrioid serous carcinoma and undifferentiated carcinoma. Their median age was 49 years (range 33–77). Ten patients had stage I disease, 8 patients had stage II disease, and 14 patients had stage III disease. Nineteen patients were HPV 16 positive, and 3 patients were HPV 18 positive. The clinical characteristics of the samples are summarized in Table 1.

### 3.2. SNVs and CNVs

In the study cohort, genomic analysis identified 810 somatic variations (including SNVs and small indels) (Figure 1); 2730 germline mutations and 701 CNVs. Somatic mutation types are summarized in Table 2. Of 810 somatic variations, the four genes with the most significant somatic mutation enrichment were PIK3CA (31.25%), MTOR (15.63%), KMT2D (12.50%) and FAT1 (12.50%), followed by MDC1 (9.38%), ANKRD11 (9.38%), APC (9.38%), BCORL1 (9.38%) and TP53 (9.38%). Overall, the number and frequency of germline variations were significantly higher than those of somatic variations (Figure 2); the mutation rate of FAT1 reached 46.88%, followed by HLA-B (40.63%) and ZFHX3 (28.13%). Multivariate analysis showed that somatic FAT1 mutations were significantly associated with nonsquamous carcinoma (*p* < 0.05); germline TET2 and SESN2 mutations were also significantly associated with nonsquamous carcinoma (both *p* < 0.05). In addition, germline PTPRT and SLX4 mutations were significantly associated with smaller tumor sizes (both *p* < 0.05). Additionally, germline BCORL1 and GPR124 were significantly associated with no HPV 16 infection and younger age, respectively (both *p* < 0.05).

Germline variations were obviously higher than those in COSMIC database, suggesting that HLA might be mutated as an antigen-presenting complex in cervical cancer. CNV analysis revealed that the PIK3CA gene loci had a particularly high frequency of CNV. Moreover, the BRCA1, BRCA2, ATM and TP53 gene loci also had a higher frequency of CNV (Figure 3).

We performed overlap analysis of the SNV genes and CNV genes and focused on 34 genes in our subsequent analysis. The protein interaction simulation analysis identified PIK3CA as the key player in the whole network; almost all other proteins directly or indirectly interacted with PIK3CA (Figure 4). Additionally, both HLA-A and HLA-B interacted with PIK3CA, which were reported to confer radioresistance to cervical cancer [6], suggesting that HLA-A and HLA-B may also be involved in radiotherapeutic resistance of cervical cancer. Furthermore, based on these 34 genes, KEGG pathway analysis revealed that virus infection-related pathways were significantly abnormal in cervical cancer (Figure 5). Additionally, GO enrichment analysis showed that those patients may have multiregulation function disorder.

### 3.3. Analysis of Oncogenic Mutations in the ONCOKB Database

To further elucidate the role of germline and somatic variations in cervical cancer, we compared the genetic variations identified in the current study with those in the ONCOKB database. Forty-four oncogenic or likely oncogenic somatic variations in thirty genes that were identified in this study are recorded for cervical cancer in the ONCOKB database, including eight oncogenic mutations and thirty-six likely oncogenic mutations (Table 3 and Appendix A). Notably, three recurrent mutations in the PIK3CA gene were located in the essential protein-coding region, such as E545K and E542K, which are reported to confer radioresistance to cervical cancer (Table 2 and Figure 6) [6]. TP53 was the second most common oncogenic mutated gene in three cases, and it is well known that TP53 inactivation is closely associated with the development of cervical cancer [15]. Additionally, FBXW7 and EP300 mutations have already been reported in cervical cancer [16]. Moreover, PIK3CA, TP53, BRCA1, ERBB3, KIT, KRAS, NRAS, PTEN and STK11 all belong to the PI3K-AKT signaling pathway, which is involved in regulating tumor growth and metastasis and radiosensitivity of cervical cancer [17]. Fifteen germline variations with pathogenicity found in thirteen genes in this study are recorded in the ONCOKB database (Table 4), suggesting that pathogenic germline variations are also implicated in the oncogenesis of cervical cancer, possibly via affecting individual cancer susceptibility [5]. In addition, we performed Metascape gene analysis on these thirteen germline mutated genes and found that almost half of these genes were enriched in the DNA double-strand break repair pathway, indicating that individual cervical cancer susceptibility may be associated with genetic DNA double-strand break repair defect.

It is well known that BRCA1/2, the vital DNA repair genes, play a particularly important role in cancer in women, such as breast cancer and ovarian cancer. However, their role in cervical cancer has not been determined. Therefore, we further conducted an in-depth analysis of BRCA1/2 and found that many samples carried multiple mutations in the coding region of BRCA1/2 (Figure 7). However, only two samples had mutations located at the important coding region of BRCA1/2, indicating that the contribution of BRCA1/2 to cervical cancer may be smaller than that in breast cancer or ovarian cancer.

### 3.4. Analysis of Potential Therapeutic Targets

Finally, we analyzed known therapeutic targets through the ONCOKB database. Ten candidate oncogenic somatic mutations were identified in 7 genes that could be immediately applicable as therapeutic targets (Table 5), including PIK3CA, ATM, BRCA1, KRAS, NRAS, PALB2 and PTEN, and the corresponding targeted drugs included fulvestrant plus alpelisib, olaparib, talazoparib, rucaparib, niraparib and others. Moreover, seven likely oncogenic germline mutations were identified in two known targeted genes, ATM and RAD51B (Table 6). Both ATM and RAD51B are important genes in the homologous recombination (HR) repair pathway, and their targeted drug olaparib has been approved by the US Food and Drug Administration (FDA) for metastatic castration-resistant prostate cancer [18], suggesting that cervical cancer patients may also benefit from PARP inhibitors, which is consistent with the enrichment results of the above germline oncogenic mutated genes.

### 3.5. Advanced Stage Cervical Cancer Exhibits Notable Chromosome 6q27 Loss

Analysis of the distribution of SNVs and SCNAs using CNVkit and GISTIC revealed significant higher rates of chromosome 6q27 loss in advanced stage versus early stage cervical cancer (Appendix A). Totally, 13 genes including HLA-A and -J, TRIM10, 15, 26, 31, 40 and others experienced SCNA loss in advanced stage cervical cancer (Table 7). Notably, GO term enrichment analysis showed that these genes were enriched in two pathways, negative regulation of viral life cycle (TRIM26, TRIM10, TRIM31, TRIM15, HLA-A and TRIM40) and regulation of cytokine production (HLA-A, PPP1R11 and TRIM15), as shown in a bubble plot, where the bubble size indicates the number of genes in corresponding cluster and the color indicates the percentage of enriched genes, with red being 100% and blue being 0% (Appendix A). Meanwhile, no significant difference in SCNA gain or amplification was observed between early stage and advanced stage cervical cancer.

## 4. Discussion

Cervical cancer is a common female malignancy with an incidence of 126.94/100,000 in China [19]. It is of overly critical importance to understand the molecular mechanisms and genetic susceptibility of cervical cancer occurrence and development for early diagnosis and clinical therapy. Ojesina et al., performed whole exome sequencing (WES) of 115 paired cervical carcinoma and normal samples, RNAseq of 79 cases, and WGS of 14 tumor-normal pairs. They revealed that squamous cell carcinomas have higher frequencies of somatic nucleotide substitutions at cytosines preceded by thymines (Tp*C sites) than adenocarcinomas. They observed previously unknown somatic mutations in the MAPK1 gene, inactivating mutations in the HLA-B gene, and mutations in EP300, FBXW7, NFE2L2, TP53 and ERBB2 in squamous cell carcinoma samples, and somatic ELF3 and CBFB mutations in adenocarcinomas. They also reported that gene expression levels at HPV integration sites were statistically significantly higher in tumors with HPV integration compared with expression of the same genes in tumors without viral integration at the same site [16]. Chung et al., performed WES in 15 Chinese cervical cancer patients. They observed frequently altered genes including FAT1, ARID1A, ERBB2 and PIK3CA. They also found HPV sequence in 13 samples and suggested that HPV genome integrated into the exon and may affect the tumorigenesis pathway [20]. One of the largest cervical cancer sequencing efforts—*The Cancer Genome Atlas (TCGA) Project*—reveals novel mutations in several genes, including SHKBP1, ERBB3, CASP8, HLA-A and TGFBR2, amplifications in immune targets PD-L1 and PD-L2, by sequencing 228 primary cervical cancer patients. They confirmed previously reported mutations in PIK3CA, EP300, FBXW7, HLA-B, PTEN, NFE2L2, ARID1A, KRAS and MAPK1. This study illuminates new therapeutic targets in cervical cancer [8]. Unfortunately, we did not find any genomic alterations that are peculiar to the Chinese population in this study, probably due to the small sample size and the limitation of the clinical samples from a single center. We are actively recruiting more samples for further investigation.

In this study, using a 571 tumor-related gene panel for NGS, we identified 810 significant somatic variations, 2730 germline mutations and 701 CNVs from 32 cervical cancer samples and paired blood samples. PIK3CA and MTOR were the most frequently mutated genes, demonstrating that the PI3K/Akt/mTOR signaling pathway is commonly activated in cervical cancer [16]. Previous studies revealed PIK3CA mutation is frequent in cervical cancer, and is associated with a poor OS and PFS [8,20,21,22,23]. PIK3CA was mutated in 14% cervical squamous cell carcinoma patients in the study by Ojesa et al., Genomic profiling of advanced cervical cancer in the CLAP trial also showed a higher PFS in women with mutated PIK3CA receiving second line or later camrelizumab plus apatinib [24]. This clinical benefit remains inconclusive in women receiving chemotherapy [25]. Scholl et al., showed that patients with altered PI3K and epigenetic pathways had significantly poorer PFS [26]. Nevertheless, these women only received conventional therapy. These findings indicate that biomarkers may have different predictive functions for cancer patients receiving conventional therapy versus immunotherapy, and a distinct set of predictive biomarkers should be developed for cervical cancer patients receiving immune checkpoint inhibitors.

Furthermore, the pathways enriched in the mutated genes in this study could offer some insight into signaling pathways associated with cervical carcinogenesis that can be therapeutically targeted such as the PI3K/Akt signaling pathway and DNA damage response pathways. DNA repair pathway genes, such as BRCA1, BRCA2 and ATM, which are potential predictive biomarkers [27,28,29], also had a high frequency of CNVs in the present study, suggesting that HR defect might serve as a therapeutic target in cervical cancer. Furthermore, we performed overlap analysis of the SNV genes and CNV genes and screened 34 genes for subsequent bioinformatics analysis. Protein interaction simulation analysis of proteins encoded by the 34 genes showed that PIK3CA occupied the crucial central position in the whole network. Additionally, the results indicated that both HLA-A and HLA-B play an important role in this network, and intimately interacted with PIK3CA. Moreover, the somatic SNV mutation rate of PIK3CA is higher in HPV 16 (37.5%) than non-HPV 16 (12.5%). However, the difference was not significant (Fisher exact test, *p* = 0.35), probably due to the limited sample size.

GO term enrichment analysis showed that the 34 genes were significantly enriched in cell cycle regulation, cellular response, and metabolic process, suggesting that some of these genes could be involved in cell cycle processes to promote cell proliferation by activating related signaling pathways and could be promising candidate genes of antitumor drugs. KEGG pathway enrichment analysis found that the virus infection-relation pathways were significantly enriched with many pathogenic genetic variations, which might contribute to persistent HPV infection.

We also observed chromosome 6q27 loss in advanced stage cervical cancer, occurring in 42% of advanced stage cervical cancer samples versus none in early stage cervical cancer samples. Chromosome 6 is frequently affected in cervical cancer [30,31]. Loss of heterozygosity (LOH) in 6q27 has been reported in up to 39% patients with invasive squamous cell carcinomas of the cervix [32]. TRIM10, 15, 26, 31 and 40 loss has not been previously described in cervical cancer and their role in carcinogenesis of the cervix remains to be defined. Most TRIM family proteins are E3 ubiquitin ligases and have been reported to be involved in carcinogenesis.

While cross checking our results with TCGA, we found that the virus infection-relation pathways were significantly enriched with many pathogenic genetic variations, which is consistent to some extent with the result in TCGA showing that the driver genes of cervical cancer in the TCGA-CESC dataset were mainly enriched in the KEGG pathway, including human T-cell leukemia virus 1 infection, human papillomavirus infection, viral carcinogenesis, human cytomegalovirus infection and PI3K-Akt signaling pathway. GO pathway in the TCGA is enriched in cellular response to abiotic stimulus, negative regulation of cell differentiation, positive regulation of cell death and negative regulation of protein modification process, whereas our GO term enrichment analysis showed that the pathway was significantly enriched in cell cycle regulation, cellular response and metabolic process. These differences are highlighted in the paper.

By comparing with the ONCOKB database, the role of PIK3CA in cervical cancer was further highlighted, especially for E545K and E542K. Furthermore, this study also identified oncogenic somatic mutations in BRCA1, CHD1, KRAS and FBXW7, and likely oncogenic somatic mutations in EP300, HLA-A, KMT2D, PTEN and TP53, among others. In addition, we observed fifteen likely oncogenic germline mutations in thirteen genes recorded in the ONCOKB database. Notably, seven patients in the present study carried the probable oncogenic germline mutation in HLA-B, a well-known cervical cancer susceptibility gene [8,16]. Moreover, we also used the ONCOKB database to identify available therapeutic targets and found that eight genes can be targeted by fulvestrant plus alpelisib, olaparib, talazoparib, rucaparib and others, highlighting the potential clinical significance of therapeutic agents targeting these mutated genes.

In this article, we validated the presence of many specific pathogenic mutations in Chinese cervical cancer patients. There are several limitations in the current study. We did not uncover new molecular targets for cervical cancer in the Chinese cervical cancer population, possible due to the number of the samples and the limitation of the clinical samples from a single center. However, we performed a comprehensive analysis and presented more details of our findings, which have been partially confirmed by some other studies. We did not investigate the relationship between HPV integration and cervical cancer due to the lack of HPV status in the patients. We will include this information in future studies. We did not investigate the association of PIK3CA mutations and survival outcomes of cervical cancer patients in our cohort. As of December 2020, except for four patients lost to follow-up, only two patients had lymph metastasis or died (one each), and the remaining patients were progression-free. In the future, we will further expand the study and continue to follow patients to evaluate the significance of these variations in patient outcomes.

## 5. Conclusions

Overall, we conducted a gene-panel sequencing analysis of 32 paired cervical cancer samples. Important gene variations and pathways were identified to provide a theoretical basis for potential drug target validation and further elucidation of the molecular mechanisms of cervical cancer. Nevertheless, additional relevant studies are needed to further evaluate their prognostic significance in cervical cancer.

## Figures and Tables

**Figure 1 genes-13-00287-f001:**
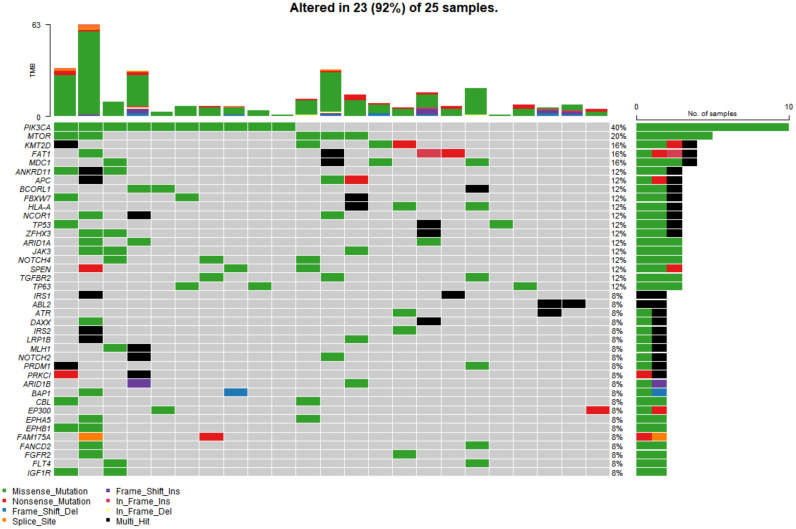
The top 40 somatic mutant genes with the highest mutation rate in 25 of 32 cervical cancer samples. Specific genetic mutations were identified by targeted next generation sequencing in the tumor tissues. The upper panel shows the number of nonsynonymous single-nucleotide variants and small insertions or deletions in each tumor.

**Figure 2 genes-13-00287-f002:**
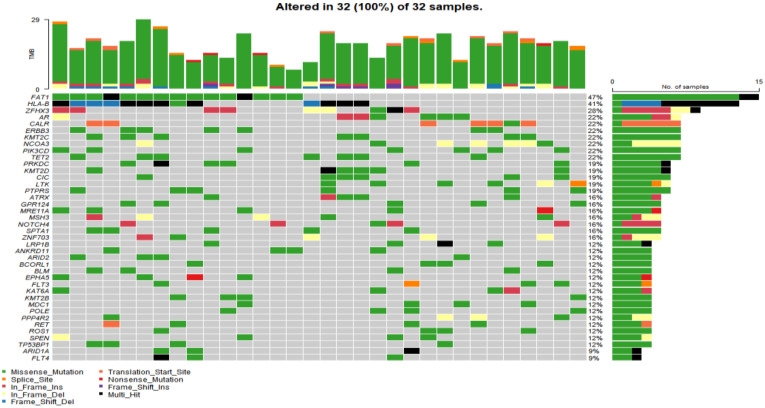
The top 40 germline mutant genes with the highest mutation rate in 32 of 32 cervical cancer samples. Genetic mutations were identified by targeted next generation sequencing in the control samples. The upper panel shows the numbers of nonsynonymous single-nucleotide variants and small insertions or deletions.

**Figure 3 genes-13-00287-f003:**
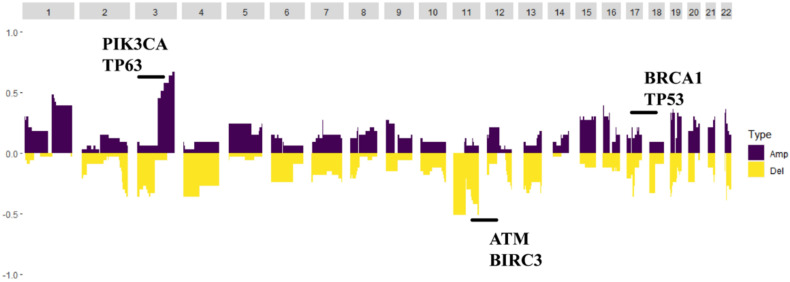
The copy number variation map; distribution shift of CNVs in the chromatins. Copy number losses (yellow) and gains (dark purple) were determined from the sequencing data. PIK3CA and TP63 were the highest frequency CNV genes, and BRCA2, TP53, ATM and BIRC3 also had a higher frequency of CNV.

**Figure 4 genes-13-00287-f004:**
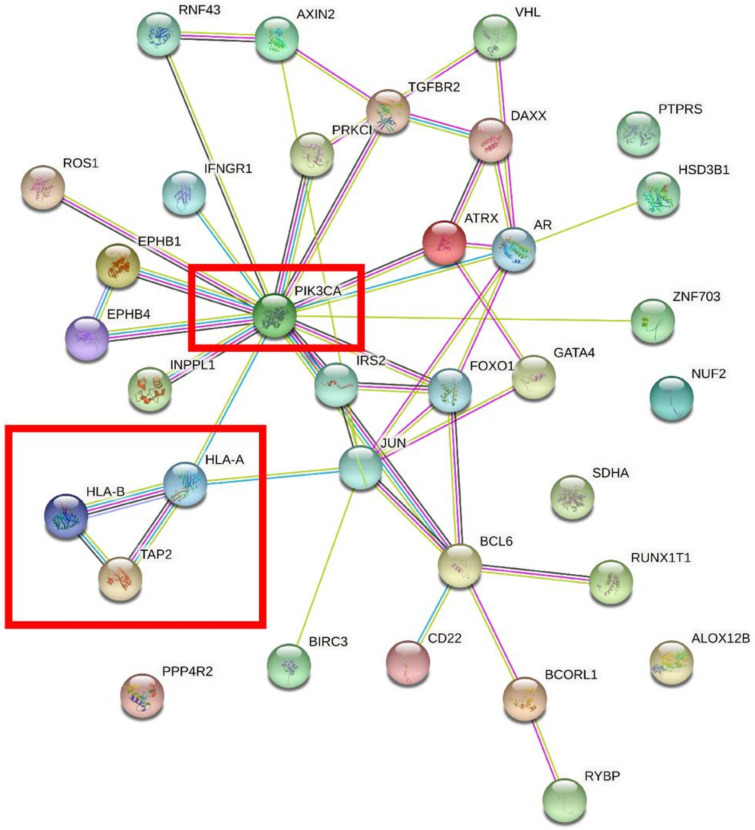
The protein interaction simulation analysis of 34 genes. PIK3CA was the key gene in the whole network.

**Figure 5 genes-13-00287-f005:**
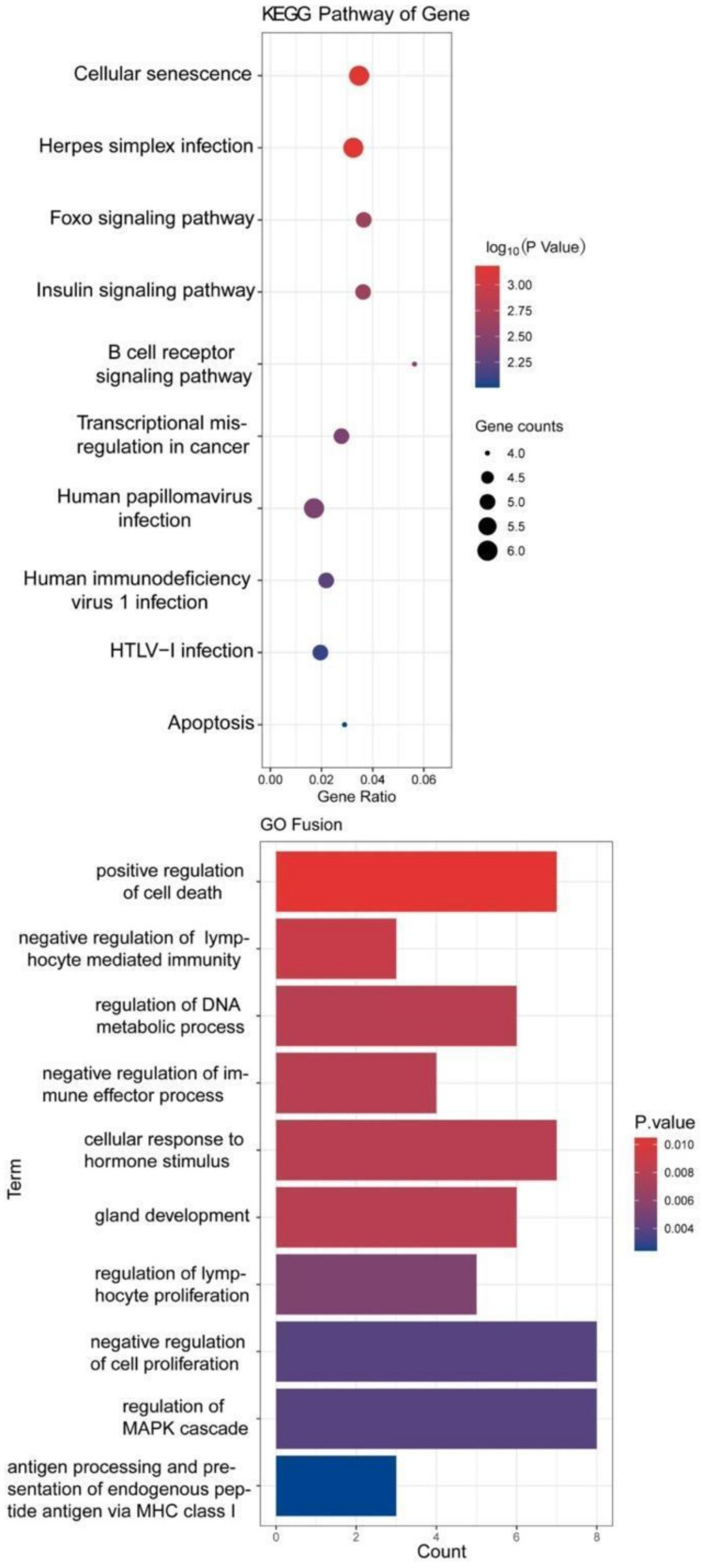
KEGG and GO analysis of 34 genes.

**Figure 6 genes-13-00287-f006:**
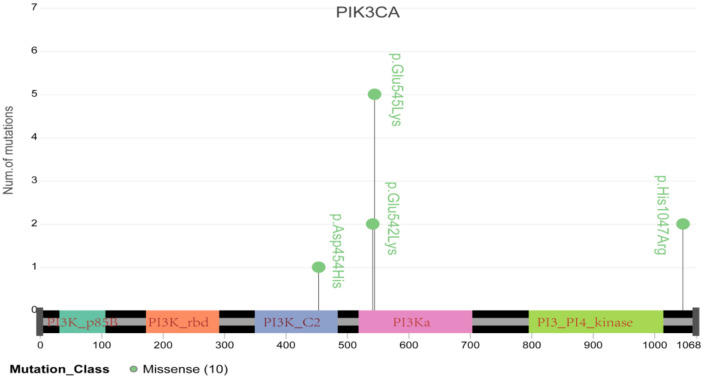
Schematic diagram of major mutations in PIK3CA, including E545K and E542K, which have been demonstrated to be tightly associated with cervical cancer resistance to radiotherapy.

**Figure 7 genes-13-00287-f007:**
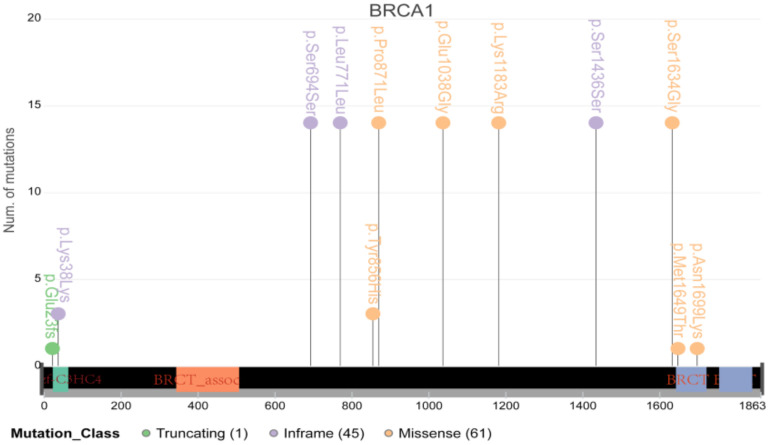
The schematic diagram of major mutations in BRCA1/2. Most mutations were located in the nonimportant coding region.

**Table 1 genes-13-00287-t001:** Demographic and baseline characteristics of the study population.

Variable	*n* = 32 (100%)
Age, years	
Median (range)	49 (33–77)
Histologic type	
Squamous carcinoma	24
Adenosquamous carcinoma	4
Adenocarcinoma	2
Endometrioid serous carcinoma	1
Undifferentiated carcinoma	1
FIGO stage	
I	10
II	8
III	14
Biomarkers	
CA125 +	4
CA199 +	2
SCCA +	6
HE4 +	1
Undetected	17
Undefined	2
HPV infection	
HPV 16	16
HPV 18	3
HPV 33	1
HPV 51	1
HPV 58	1
High risk	2
Negative	1
Undefined	7

**Table 2 genes-13-00287-t002:** Summary of somatic mutations.

Variant Classification	Count
Intron	263
Missense Mutation	250
Silent	113
Upstream Gene Variant	41
Downstream Gene Variant	24
Splice Site	29
Nonsense Mutation	23
3’UTR	17
Frame Shift Ins	14
Frame Shift Del	10
Intragenic Variant	8
5’UTR	6
In Frame Del	4
In Frame Ins	4
Translation Start Site	4

**Table 3 genes-13-00287-t003:** Oncogenic somatic variations identified in this study and labeled in the ONCOGENIC database.

Genes	Variant_Classification	HGVS.c	HGVS.p	Sample Count	Origin
PIK3CA	Missense_Mutation	c.1633G>A	p.Glu545Lys	5	Somatic
PIK3CA	Missense_Mutation	c.3140A>G	p.His1047Arg	2	Somatic
PIK3CA	Missense_Mutation	c.1624G>A	p.Glu542Lys	2	Somatic
BRCA1	Frame_Shift_Ins	c.66dupA	p.Glu23fs	1	Somatic
CDH1	Missense_Mutation	c.1018A>G	p.Thr340Ala	1	Somatic
KRAS	Missense_Mutation	c.35G>A	p.Gly12Asp	1	Somatic
NRAS	Missense_Mutation	c.35G>A	p.Gly12Asp	1	Somatic
FBXW7	Missense_Mutation	c.1393C>T	p.Arg465Cys	1	Somatic

**Table 4 genes-13-00287-t004:** Likely oncogenic germline variations identified in this study and labeled in the ONCOGENIC database.

Gene	Variant_Classification	HGVS.c	HGVS.p	Sample Count	Origin
HLA-B	Frame_Shift_Del	c.354_355delCC	p.Leu119fs	7	Germline
PPP6C	Frame_Shift_Ins	c.152dupC	p.Pro52fs	3	Germline
MUTYH	Splice_Site	c.934-2A>G		2	Germline
AXIN1	Translation_Start_Site	c.-135C>T		1	Germline
CASP8	Translation_Start_Site	c.-30T>A		1	Germline
MRE11A	Nonsense_Mutation	c.1447C>T	p.Arg486Ter	1	Germline
PIK3R2	Splice_Site	c.901+1G>A		1	Germline
RAD50	Frame_Shift_Ins	c.2165_2166insT	p.Lys722fs	1	Germline
RAD51B	Splice_Site	c.316-4_316-3dupTT		1	Germline
RAD51B	Splice_Site	c.316-5_316-3dupTTT		1	Germline
RECQL	Translation_Start_Site	c.2T>C	p.Met1Thr	1	Germline
RECQL4	Nonsense_Mutation	c.3328G>T	p.Glu1110Ter	1	Germline
TP53	Missense_Mutation	c.790C>G	p.Leu264Val	1	Germline
ZFHX3	Frame_Shift_Ins	c.9583_9584insT	p.Pro3195fs	1	Germline
ZFHX3	Frame_Shift_Ins	c.9588_9589insAG	p.Gln3197fs	1	Germline

**Table 5 genes-13-00287-t005:** Candidate oncogenic somatic mutations in the targeted genes.

Gene	HGVS.c	HGVS.p	Samples	Mutation Effect	Oncogenic	Targeted Drugs	Origin
PIK3CA	c.1633G>A	p.Glu545Lys	5	Gain-of-function	Oncogenic	Fulvestrant + Alpelisib	Somatic
PIK3CA	c.3140A>G	p.His1047Arg	2	Gain-of-function	Oncogenic	Fulvestrant + Alpelisib	Somatic
PIK3CA	c.1624G>A	p.Glu542Lys	2	Gain-of-function	Oncogenic	Fulvestrant + Alpelisib	Somatic
ATM	c.1899-7C>A	--	1	Likely loss-of-function	Likely Oncogenic	Olaparib	Somatic
BRCA1	c.66dupA	p.Glu23fs	1	Loss-of-function	Oncogenic	Olaparib, Talazoparib, Rucaparib, Niraparib	Somatic
KRAS	c.35G>A	p.Gly12Asp	1	Gain-of-function	Oncogenic	Trametinib, Cobimetinib, Binimetinib	Somatic
NRAS	c.35G>A	p.Gly12Asp	1	Gain-of-function	Oncogenic	Binimetinib	Somatic
PALB2	c.3477G>A	p.Trp1159 Ter	1	Likely loss-of-function	Likely Oncogenic	Olaparib	Somatic
PTEN	c.469G>T	p.Glu157Ter	1	Likely loss-of-function	Likely Oncogenic	GSK2636771, AZD8186	Somatic
PTEN	c.640C>T	p.Gln214Ter	1	Likely loss-of-function	Likely Oncogenic	GSK2636771, AZD8186	Somatic

**Table 6 genes-13-00287-t006:** The likely oncogenic germline mutations in the targeted genes.

Gene	HGVS.c	HGVS.p	Samples	Mutation Effect	Oncogenic	Targeted Drugs	Origin
ATM	c.3154-5C>T	--	1	Likely loss-of-function	Likely	Olaparib	Germline
RAD51B	c.316-4_316-3dupTT	--	1	Likely loss-of-function	Likely	Olaparib	Germline
RAD51B	c.316-5_316-3dupTTT	--	1	Likely loss-of-function	Likely	Olaparib	Germline
RAD51B	c.316-3delT	--	2	Likely loss-of-function	Likely	Olaparib	Germline
RAD51B	c.316-4_316-3delTT	--	2	Likely loss-of-function	Likely	Olaparib	Germline
RAD51B	c.316-6_316-3delTTTT	--	1	Likely loss-of-function	Likely	Olaparib	Germline
RAD51B	c.316-18_316-3delTTTTTTTTTTTTTTTT	--	1	Likely loss-of-function	Likely	Olaparib	Germline

**Table 7 genes-13-00287-t007:** SCNA loss in chromosome 6q27 in early stage vs. advanced stage cervical cancer.

Genes	Early Stage Cervical Cancer (+/−)	Advanced Stage Cervical Cancer (+/−)	*p*-Value	Adjusted *Q* Value
HLA-A	0/10	5/7	0.040	1
HLA-J	0/10	5/7	0.040	1
PPP1R11	0/10	5/7	0.040	1
TRIM26	0/10	5/7	0.040	1
TRIM10	0/10	5/7	0.040	1
HCG9	0/10	5/7	0.040	1
TRIM31	0/10	5/7	0.040	1
ZNRD1	0/10	5/7	0.040	1
RNF39	0/10	5/7	0.040	1
ZNRD1-AS1	0/10	5/7	0.040	1
TRIM15	0/10	5/7	0.040	1
TRIM40	0/10	5/7	0.040	1
HCG8	0/10	5/7	0.040	1

+: positive for SCNA loss or deletion according to GISTIC status; −: negative for SCNA loss or deletion.

## Data Availability

The authors confirm that the data supporting the findings of this study are available within the article. Raw data that support the findings of this study are available from the corresponding authors, upon reasonable request.

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
