# Peer review of "Characterization of the Genomic Landscape in Cervical Cancer by Next Generation Sequencing"

_genes, 2022, doi:10.3390/genes13020287_

Round 1

Reviewer 1 Report

The authors analyzed 64 samples, consisting of 32 tumours and 32 blood samples 12 from 32 Chinese cervical cancer patients by multigene targeted NGS.

The study is not novel as there are available many studies with WES or WGS but is interesting that has been performed in a homogenous population (Chinese) and in a short period of time.

The manuscript is well written and the results are well presented.

Just a few minor points.

1) The authors should present when the biopsies were taken? during colposcopy or during primary surgery? Were all primary cases or some were recurrent? These data should be presented better.

2) Which is the depth of sequencing? Are these values used to filtered false-positive applicable to clinical implementation?

3) The authors should mention in the results and discussion which are the genes/pathway similar to the TGCA and which they think might be peculiar to the Chinese population.

4) One paragraph is called "mutation burden and mutational signatures" the title is misleading as mutational burden or TMB which is possible to define from targeted sequencing it is an important parameter to potential response to immune checkpoint blockade.

Mutational signatures are well-described signatures derived from WES. None of the two analyses was present in the paragraph mentioned. Please insert the analysis of TMB from your NGS data or change the title of the paragraph

Reviewer 2 Report

This is a well-written paper by Ling Qiu and colleagues presenting the results of next generation analysis of 64 samples from cervical cancer patients and control subjects.

Multigene NGS analysis comprised 571 cancer-related genes. The authors performed Somatic Variant Identification, Germline Variant Identification, Copy Number Variations (CNVs) analysis, and Gene Ontology (GO) and KEGG Pathway enrichment analyses. In this study, using a 571 tumor-related gene panel for NGS, authors identified 810 signifi-298 cant somatic variations, 2730 germline mutations and 701 CNVs from 32 cervical cancer 299 samples and paired blood samples. PIK3CA and MTOR were the most frequently mutated genes, demonstrating that the PI3K/Akt/mTOR signaling pathway is commonly activated in cervical cancer. Protein interaction simulation analysis of proteins encoded by the 34 genes showed that PIK3CA occupied the crucial central position in the whole network. Both HLA-A and HLA-B interacted with PIK3CA, suggesting that HLA-A and HLA-B may also be involved in radiotherapeutic resistance of cervical cancer. DNA repair pathway genes, BRCA1, BRCA2 and ATM, which are potential predictive biomarkers had a high frequency of CNVs, suggesting that HR defect might serve as a therapeutic target in cervical cancer. This study also identified 342 oncogenic somatic mutations in BRCA1, CHD1, KRAS, and FBXW7, and likely oncogenic somatic mutations in EP300, HLA-A, KMT2D, PTEN, TP53. The authors documented  loss of chromosome 6q27 and associated genes: TRIM10, 15, 26, 31 and 40 in 42% of cervical cancer samples.

Presentation of the results in the manuscript is easy to follow for readers. The results of above analyses were showed in 8 figures and 9 tables. The authors extensively discussed their results and latest literature data. Furthermore, the authors report possible limitation of their study that might lead to risk of some biased interpretations. Paper has 30 reherences which are relevant to article's subject. This is an excellent study, and clinically valuable, especially for those researchers who seek comprehensive data on genetic susceptibility of cervical cancer, and its related resistance to radiotherapy. This manuscript provide comprehensive information on this issue.

Minor comments:

Comment 1. HPV status of the cervical cancer patients was described and listed in Table 1 and Supplementary Table 3. It would be beneficial for readers to see if there are any correlations between HPV type and particular genetic instability events.

Comment 2. Figure 3. In my eyes, copy number loss is illustrated as a blue area while in figure legend it is described as green.

Taken together, this paper by Ling Qiu and colleagues represents a worthwhile contribution to the cancer research. I recommend the manuscript for further publication process.

Reviewer 3 Report

The Manuscript “Characterization of the genomic landscape and therapeutic targets in cervical cancer by next generation sequencing” will be a good addition in the existing literature of cervical cancer. The manuscript is written well and the reasonable conclusions are drawn. Regarding the samples it would have been better to have non-cancerous cervical samples rather than the bold sample, but since the focus of the study was to observe the genomic alterations, I believe the blood sample is good enough to serve the purpose. Overall, the manuscript is in good condition.

For the betterment I will suggest –

  1. Please work on references and the referencing style.
  2. The introduction part is written well, but at some places the statics are confusing, like page 2, line 42/43– “cervical cancer is the most frequent cancer in women in 28 countries and the leading cause of cancer death in 48 countries, which are mostly located in sub-Saharan Africa and South-Eastern Asia” needs better representation.
  3. The manuscript does Characterization of the genomic landscape in cervical cancer, and does provide the altered genes in these patients but the “therapeutic targets” parts as mentioned in the title is not present in the manuscript, even how these altered genomic loci can be used as therapeutic targets is not well discussed, I will suggest either revise the discussion part with some experimental and/or literature proves to support this part or rethink the title of the manuscript.

Reviewer 4 Report

There are several inaccurate argument  manuscript.

64 samples, consisting of 32 tumors and 32 blood samples--Small population since TCGA

810 somatic variants, 2730 germline mutations and 701 copy number 14 variations (CNVs) --Too many genes so far since TCGA were found. In clinical setting and trial, we need actionable, more accurate target gene not just related genes.

Cervical cancer patients could benefit from PARP inhibitors-->Synthetic lethality has not proven in cervical cancer regarding PARP inhibitor in cervical cancer. Not realistic so far

Genetic susceptibility of cervical cancer, but also influence the resistance of cervical cancer to radio

therapy-->HPV carcinogenesis is predominant in cervical cancer.Germline mutation is not much related. Sample groups are not classified radiosensitive vs radioresistant group.

Round 2

Reviewer 4 Report

Originality & Novelty are still not appealing